

# Resection vs. ablation for lesions characterized as resectable-ablative within the colorectal liver oligometastases criteria: a propensity score matching from retrospective study

Ma Luo[1,*], Si-Liang Chen[1,*], Jiawen Chen[1], Huzheng Yan[1], Zhenkang Qiu[1], Guanyu Chen[1], Ligong Lu[2] and Fujun Zhang[1]

[1] Sun Yat-sen University Cancer Center; State Key Laboratory of Oncology in South China; Collaborative Innovation Center for Cancer Medicine, Guangzhou, China
[2] Zhuhai Interventional Medical Center, Zhuhai Precision Medical Center, Zhuhai People's Hospital, Zhuhai Hospital Affiliated with Jinan University, Zhuhai, Guangdong, China
[*] These authors contributed equally to this work.

Corresponding authors
Ligong Lu, luligong1969@126.com
Fujun Zhang, zhangfj@sysucc.org.cn

## ABSTRACT

**Background.** There has been no prospective or retrospective studies reporting the comparison outcome between surgery and ablation for resectable-ablative (lesions could be treated by resection or complete ablation) colorectal liver oligometastases (CLOM). The purpose of this study was to compare the efficacy and prognostic difference in patients who underwent R0 resection vs. complete ablation within the resectable-ablative CLOM criteria.

**Methods.** From January 2008 to May 2018, a total of 2,367 patients diagnosed with colorectal liver metastases were included in this observational study. The metastasis was characterized by only limited to liver with number $\leq 5$, size $\leq 5$ cm, and resectable-ablative (lesions could be treated by resection or complete ablation). The evaluated indications, including liver progression-free survival (LPFS), overall survival (OS), survival rates, pattern and number of recurrences, and complications, were compared by using propensity score matching (PSM). The Kaplan–Meier curves were generated, and a log-rank test was performed. The Cox regression model was used for univariate and multivariate analyses to identify predictors of outcomes.

**Results.** A total of 421 consecutive patients were eligible for this study, with 250 and 171 undergoing R0 resection and complete ablation, respectively. PSM identified 145 patients from each group. The 1-, 3-, 5- and 8-year OS rates in the resection group and the ablation group were 95.8% vs. 95.0%, 69.8% vs. 60.1%, 53.6% vs. 42.5%, and 45.1% vs. 32.9% ($p = 0.075$), respectively. The median LPFS in the resection group was significantly longer than that in the ablation group (35 months vs. 15 months, $p = 0.011$). No statistical difference was found in LPFS between the two groups when comparing $\leq 3$ cm liver metastases. For liver metastasis $>3$ cm, the median LPFS in the resection group and ablation group was 11 months and 5 months, respectively ($p = 0.001$). In terms of high risk of clinical risk score (CRS), the resection group showed longer LPFS than the ablation group (median 18 months vs. 10 months, $p = 0.043$).

**Conclusion.** For patients within the CLOM criteria suggesting that liver metastases were resectable as well as ablative, resection could result in longer liver recurrence-free

survival than ablation in cases with size >3 cm or high risk of CRS. But for ≤3 cm liver metastases, their treatment efficacies were comparable.

# INTRODUCTION

Colorectal cancer is the third leading cause of cancer deaths in both males and females all over the world (*Siegel, Miller & Jemal, 2019*). Liver is the most frequent metastatic site, and it is estimated that about quarter of patients present with synchronous metastases when they are initially diagnosed and almost 50% of patients will develop metachronous metastases afterwards (*Van Cutsem et al., 2014*). Over the past two decades, the clinical outcome of colorectal liver metastases has improved dramatically due to advancement in the treatment regimen and alteration for treatment conception (*Van Cutsem et al., 2016*).

Colorectal liver oligometastases (CLOM) is a stage of disease characterized as the existence of limited metastases at less than three sites and five or occasionally more lesions. A local treatment approach can be used with a view to improving disease control and prognosis due to its localized features. The current regimen is resection and local ablative treatment. Resection is the most widely used option in liver metastasis and is the most common evidence-based approach, with the 5- and 10-year survival rate and the recurrence rate was 34% −58%, 25% −50%, and 11%, respectively (*Abdalla et al., 2004*; *Chua et al., 2011*; *Lin et al., 2018*). Ablation, one of approaches in the toolbox of local ablative treatment, is a type of minimally invasive treatment that is generally accepted in a wider range of patients with increasing used frequency due to its repeatability, effectiveness, and safety. In the literature concerning ablation, it was reported that the 5- and 10-year survival rate and the tumor control outcome was 31%–47.8%, 18%, and 32.3%–48%, respectively, in these patients (*Abdalla et al., 2004*; *Shady et al., 2016*; *Solbiati et al., 2012*).

However, to date, there has been no prospective study reporting the comparison outcome between the ablation and surgery for this liver disease (*Cirocchi et al., 2012*). All retrospective studies performed comparison between resection/ablation combinations or alone. Among the studies comparing the result between resection and ablation, favorable efficacy in the former group was reported because the lesion in the latter group was either unresectable, the patient was in a poor condition, or there was inequality in the number of cases (*Hur et al., 2009*; *Lee et al., 2008*; *Otto et al., 2010*; *Van Tilborg et al., 2011*). Thus, the finding of a better outcome in the resection group might be biased although the basic characteristics were balanced between the two groups. In an ablation study, *Evrard et al. (2012)* reported a survival benefit of ablation treatment in CLOM patients, but the lesions were unresectable, multiple (<13), and some of which were recurrent. These conditions could lead to bias in the results and could weaken the conclusion of these studies (*Dupre et al., 2017*; *Sofocleous et al., 2011*). From a technical aspect, if liver metastases from colorectal cancer were resectable-ablative (indicating either resection or complete ablation could be

made to eradicate such lesion), was there any difference or what was the difference in the prognostic outcome between patients treated with resection and ablation? The clinical evidence of this comparison is lacking (*Wang et al., 2018*). Therefore, we conducted this study to investigate the survival and treatment efficacy of resection versus ablation in such two groups from a cohort of CLOM patients.

## MATERIALS AND METHODS

All procedures performed involving human participants in this study were in accordance with the Declaration of Helsinki (1975) and its later amendments or comparable ethical standards. Institutional Review Board approval was obtained from the independent ethics committee at out center (approval number: 20170302015). Informed consent was obtained from all patients included in this study.

### Patient selection and data collection

A total of 2367 consecutive patients diagnosed with colorectal liver metastases from January 2008 to May 2018 at our center were retrospectively reviewed.

The inclusion criteria were as follows: (1) R0 resection for adenocarcinomas of the colon and rectum; (2) >18 years of age, Eastern Cooperative Oncology Group (ECOG) <2; (3) before enrollment, only liver oligometastases (number ≤5, size ≤5 cm). In patients who received chemotherapy before enrollment, the number of liver metastases should also be limited ≤5; (4) no previous hepatectomy or ablation for liver metastases; (5) before liver resection or ablation, the lesion was technically considered to be resectable-ablative that was qualified to undergo complete both resection and ablation consensus from one interventional radiologist and one hepatobiliary surgeon by reviewing the patient's initial CT or MRI images (Fig. 1). The assessment was performed independently by them who were blinded to the clinical information of the patients. Any disagreement on the assessment result was resolved by final consensus; (6) the liver treatment was implemented successfully. For resection, technical success was defined as R0 resection; for ablation, technical success was referred to the lesion which was completely covered by the ablation defect at the first month of follow-up images according to contrast-enhanced CT or MRI.

The exclusion criteria were as follows: (a) pathological examination was non-adenocarcinoma, such as neuroendocrine carcinoma or gastrointestinal stromal tumor; (b) no detailed medical records or data that could be further analyzed, or loss to follow-up; (c) the regimen was chemotherapy or radiotherapy alone or in combination, or hepatectomy + ablation concurrently; (d) extrahepatic metastases or >5 liver metastases at initial diagnosis; (e) liver metastases without R0 resection or complete ablation; (f) the liver lesion was in Segment I, or it was adjacent to the heart, gall bladder, gastrointestinal tract, diaphragm, or large vessel structure within one cm (Fig. 1).

The flowchart of patient selection is shown in Fig. 2.

### Ablation treatment

Ablation was implemented by MWA or RFA. MWA was performed by FORSEA MTC-3C microwave system (Qinghai Microwave Electronic Institute, Nanjing, China) with 2450

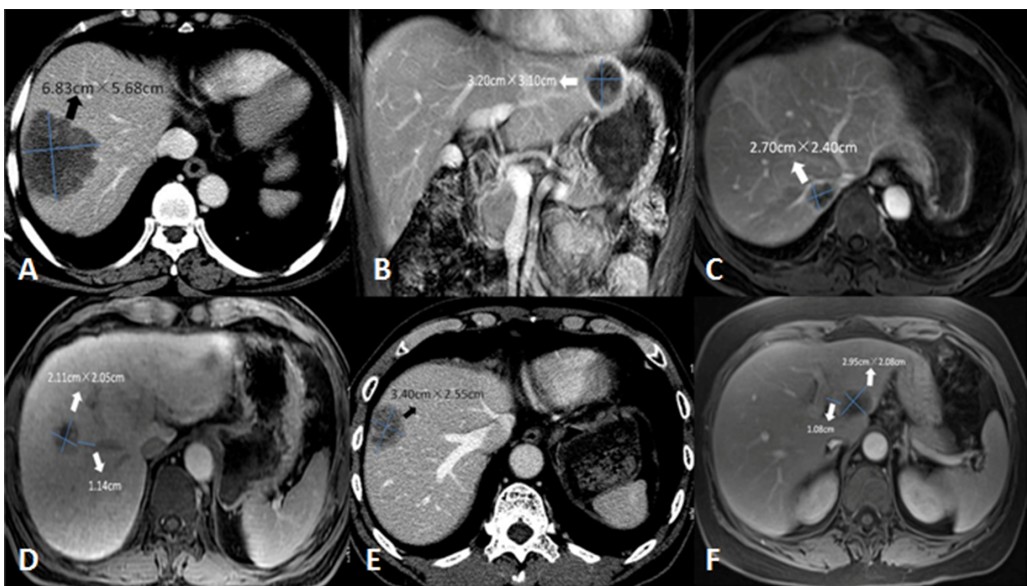

**Figure 1 Interpretation of lesion with resectable-ablative characteristics.** Liver metastasis can be categorized as resectable but not ablative (A–C), unresectable nor unablative from a technical point of view, and resectable-ablative (resection or complete ablation) (D–F) based on tumor size, location or distance with surrounding vital structure. (A–B) These two lesions (6.8 × 5.7 cm and 3.2 × 3.1 cm) were in Segment VII/VIII and the left lobe that was adjacent to the heart and stomach, respectively. They were suitable for resection but not for ablation due to their large size or proximity to the adjacent organ, which was high risk for tumor residual and complications. (C) The lesion (2.7 × 2.4 cm) was located in Segment VII, compressing the right hepatic vein and the inferior vena cava. This lesion was suitable for resection but not for complete ablation due to its poor anatomic location resulting in highly recurrence. (D–F) These lesions, classified as resectable-ablative, could be treated by resection or complete ablation without the limit of the size, the location (beyond more than 1 cm from peripheral large vessel such as hepatic vein or portal vein) or difficulty from the procedure technique after obtaining consensus of a radiologist and a hepato-biliary surgeon.

MHz frequency and a mean power of 65 W (range, 50–100 W) for 3–12 min per ablation according to the location and the size of lesions. RFA was performed by the radiofrequency system (RF 2000; RadioTherapeutics, Mountain View, CA, USA), with a mean power of 110 W (range, 40–200 W) for 4–15 min each procedure.

Contrast enhancement was routinely performed after the ablation immediately. Repeat ablation was performed when a residual tumor or a suspicious area was detected on imaging. After accomplishing complete ablation that the lesion was covered completely by the ablation defect on real-time imaging detection, the needle path was ablated to avoid bleeding or seeding.

## Follow-up and efficacy analysis

The follow-up period was the interval from the date of treatment to death or the last visit by June 1, 2019. According to recurrence patterns, the first site of recurrence was classified as local tumor recurrence that was defined as any new peripheral or nodular enhancement within one cm outside the area of previously treated field; liver new metastasis was categorized as any recurrence remote from the treated site; and extrahepatic recurrence
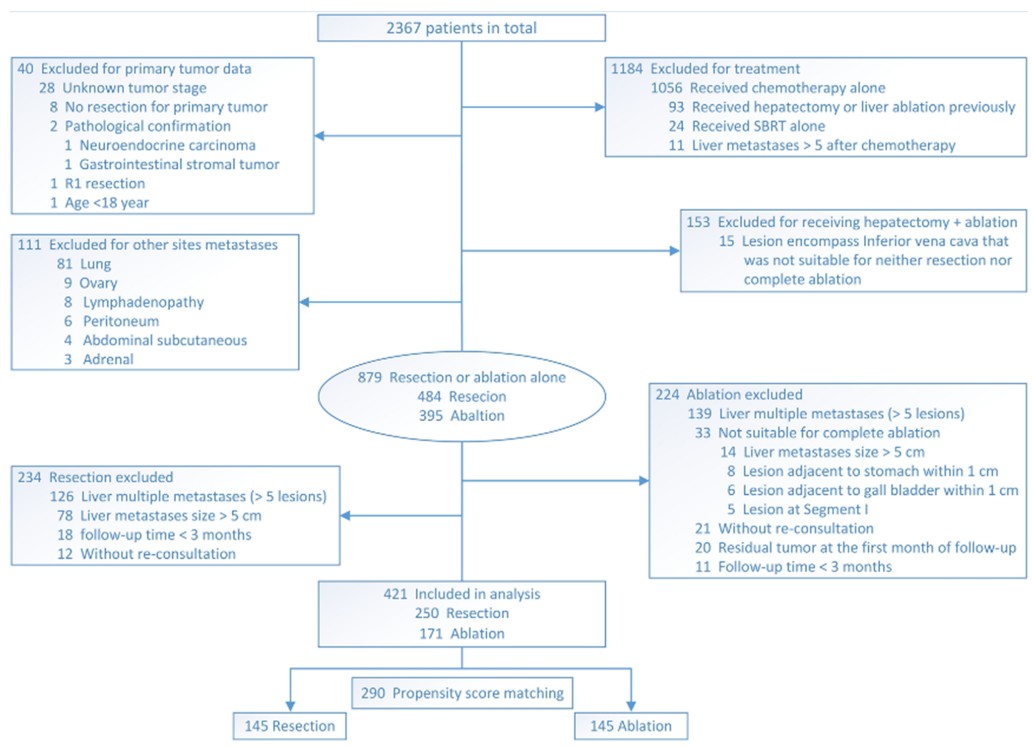

**Figure 2  Flowchart of selection of patients who underwent resection or ablation.**

was indicated as any other recurrence outside the liver based on CT or MRI enhancement during follow-up (*Stattner et al., 2013*).

The primary endpoint was liver progression-free survival (LPFS, which was defined as the duration from treatment to the first event: liver recurrence, death or the last follow-up). The secondary endpoints were overall survival (OS, which was defined as the duration from treatment to the event: death or the last visit), local tumor recurrence rate and pattern, and complications. These indications were compared between the two groups.

## Statistical analysis

To balance the basic condition and to minimize the confounding on selection bias before analysis, propensity score matching (PSM) was used and estimated by a logistic regression model using the following independent variables: age, sex, primary tumor lymph node status, disease-free interval, the number and size of liver metastases, and chemotherapy after resection or ablation. A 1:1 matching ratio between the two groups was set using the nearest-neighbor matching method (caliper = 0.1).

Statistical analysis was performed by IBM SPSS version 22.0 software (SPSS, Chicago, IL, USA) and GraphPad Prism version 6.01 (GraphPad Software, Inc., USA). The results were compared by $\chi^2$ tests, Student t test, or Fisher's exact test as appropriate.

The survival curves were calculated and compared by using the Kaplan–Meier method and the log-rank test. Univariate analysis was performed, and any factors showing statistical

significance were analyzed by the multivariate Cox hazards model. $P$ value <0.05 was considered significance (two-sided).

## RESULTS

### Patient characteristics

The baseline characteristics before PSM are summarized in Table 1. Metastases diameter and chemotherapy after the procedure were significantly different between the two groups ($P < 0.05$), and these two indices were crucial to the prognosis. After PSM, there were 145 patients in each of the resection and ablation group. Among the selected patients, there was no significant difference in any of the variables listed in Table 2.

### Treatment and local control evaluation

All enrolled patients in the resection group underwent R0 resection. There were 277 lesions in 145 patients, and the mean number of lesions treated per patient was 1.91. In the ablation group, complete ablation was achieved in all enrolled patients and no enhancement lesion was detected within the ablation area at the first month of follow-up imaging. A total of 273 ablation sessions were performed in 145 patients, and the mean number was 1.88 per patient. The local tumor recurrence rate in the resection group and the ablation group was 46.20% and 60.69% ($p = 0.013$), respectively. The local tumor recurrence rate in the MWA group and the RFA group was 36.58% and 46.03% per patient, respectively, without significant difference ($p = 0.251$), and this rate between these two groups was 31.64% and 26.96% per lesion ($p = 0.402$), respectively. The detailed recurrence pattern and number are listed in Fig. 3 and Table 3.

### Survival related outcome

The median follow-up period and OS in all patients was 32 months (range, 1–133 months) and 60 months, respectively. The entire 1-, 3-, 5-, 8- and 10-year OS rate was 95.4%, 65.3%, 48.4%, 39.4% and 32.5%, respectively (The survival and recurrence rate at specific year in the two groups are shown in Supplementary information Table 4). The median OS was 62 months (range, 1–133 months) and 53 months (range, 3–113 months) in the resection and ablation group, respectively ($p = 0.075$; Figs. 4A and 4B). Before PSM, the median LPFS in the resection group was longer than that in the ablation group (28 months vs. 12 months, $p = 0.03$). After PSM, the median LPFS was 35 months in the resection group, which was significantly longer than that of 15 months in the ablation group ($p = 0.011$; Figs. 4C and 4D).

### Subgroup analysis

There was no statistical difference in LPFS between the two groups when comparing ≤3 cm solitary or multiple liver metastases. For liver metastasis >3 cm, the median LPFS in the resection and ablation group was 11 months and 5 months ($p = 0.001$), respectively (Figs. 5A–5E).

The LPFS in patients with intravascular tumor thrombus (IVTT) negative for primary tumor was significantly longer than that with IVTT positive (35 months vs. 14 months,

**Table 1  Baseline characteristics of patients treated with resection or ablation before PSM.**

| Characteristics | Resection ($n = 250$, %) | Ablation ($n = 171$, %) | P value |
|---|---|---|---|
| Age | | | |
| Average | $55.68 \pm 10.64$ | $56.58 \pm 13.15$ | 0.385 |
| ≤60 | 161(64.4) | 103(60.2) | |
| >60 | 89(35.6) | 68(39.8) | |
| Gender | | | 0.285 |
| Male | 163(65.2) | 120(70.2) | |
| Female | 87(34.8) | 51(29.8) | |
| Comorbidity | | | 0.766 |
| No | 144(57.6) | 96(56.1) | |
| Yes | 106(42.4) | 75(43.9) | |
| Primary tumor side | | | 0.914 |
| Left-side colorectum | 197(78.8) | 134(78.4) | |
| Right-side colon | 53(21.2) | 37(21.6) | |
| TNM stage | | | |
| T stage | | | 0.067 |
| 1–3 | 122(48.8) | 68(39.8) | |
| 4 | 128(51.2) | 103(60.2) | |
| N stage | | | 0.183 |
| 0 | 113(45.2) | 64(37.4) | |
| 1 | 96(38.4) | 69(40.4) | |
| 2 | 41(16.4) | 38(22.2) | |
| IVTT[a] | | | 0.734 |
| Positive | 89(35.6) | 51(29.8) | |
| Negative | 126(50.4) | 78(45.6) | |
| Differentiation (primary) | | | 0.446 |
| Poor | 58(23.2) | 35(20.5) | |
| Moderate | 181(72.4) | 124(72.5) | |
| Well | 11(4.4) | 12(7.0) | |
| Metastases diameter (cm) | | | 0.001 |
| ≤3 | 189(75.6) | 152(88.9) | |
| >3, ≤5 | 61(24.4) | 19(11.1) | |
| Metastases number | | | 0.561 |
| 1 | 126(50.4) | 88(51.5) | |
| 2 | 56(22.4) | 45(26.3) | |
| 3 | 33(13.2) | 16(9.3) | |
| 4–5 | 35(14.0) | 22(12.9) | |
| Metastases distribution | | | 0.921 |
| Unilobar | 137(54.8) | 95(55.6) | |
| Multilobar (>1 lobe) | 113(45.2) | 76(44.4) | |
| Preoperative chemotherapy | | | 0.855 |
| No | 84(33.6) | 56(32.7) | |
| Yes | 166(66.4) | 115(67.3) | |

**Table 1** (*continued*)

| Characteristics | Resection ($n = 250$, %) | Ablation ($n = 171$, %) | P value |
|---|---|---|---|
| Postoperative chemotherapy | | | <0.001 |
| No | 22(8.8) | 42(24.6) | |
| Yes | 228(91.2) | 129(75.4) | |
| KRAS status[b] | | | 0.391 |
| Wild type | 71(28.4) | 49(28.7) | |
| Mutation type | 42(16.8) | 22(12.9) | |
| CEA level (ng/ml) | | | 0.899 |
| ≤5 | 111(44.4) | 77(45.0) | |
| >5 | 139(55.6) | 94(55.0) | |
| Timing of metastasis (months) | | | 0.084 |
| Synchronous (<12) | 29(11.6) | 30(17.5) | |
| Metachronous (≥12) | 221(88.4) | 141(82.5) | |

**Notes.**
[a] Data of 344 patients were available.
[b] Data of 184 patients were available.
PSM, Propensity score matching; IVTT, Intravascular tumor thrombus.

$p = 0.011$). In the subgroup analysis of IVTT positive and negative, respectively, although the median LPFS in the resection group was longer than that in the ablation group, there was no statistical difference (18 months vs. 12 months, $p = 0.298$; 53 months vs. 23 months, $p = 0.128$, respectively).

The LPFS was significantly shortened in patients with high risk (3 to 5) of clinical risk score (CRS, including the primary tumor node state, CEA level, size and number of liver metastases, and synchronous or metachronous metastases) compared with those with low risk (0 to 2)(40 months vs. 12 months, $p < 0.001$). In the subgroup analysis of high risk of CRS, the resection group showed longer LPFS than the ablation group (18 months vs. 10 months, $p = 0.043$, Figs. 6A and 6B).

For patients that the liver was not the first site of recurrence had longer survival duration than those with the liver was the first site of recurrence (64 months vs. 53 months, $p = 0.032$). Likewise, in the subgroup analysis of liver recurrence, patients with early recurrence (occurred within 6 months after liver curative treatment) showed shorter survival outcome than those without early recurrence (19 months vs. 40 months, $p = 0.001$, Figs. 6C and 6D).

## Survival association analysis

The univariate analysis showed that T4 stage, N stage positive, metastases diameter >3 cm, metastases number >1, high risk of CRS, ablation treatment, and absence of post-procedure chemotherapy were correlated with shorter LPFS. In the multivariate model, T4 stage (HR 1.511; $P = 0.033$), metastases diameter >3 cm (HR 2.174; $P = 0.007$) and high risk of CRS (HR 1.706; $P = 0.007$) were identified as prognostic predictors of shorter LPFS (Supplementary information Table 5). The forest plot analysis of factors associated with LPFS is shown in Fig. 7.

**Table 2   Baseline characteristics of patients treated with resection or ablation after PSM.**

| Characteristics | Resection ($n = 145$, %) | Ablation ($n = 145$, %) | P value |
|---|---|---|---|
| Age | | | |
| Average | 57.15 ± 10.26 | 56.04 ± 13.22 | 0.629 |
| ≤60 | 87(60.0) | 91(62.8) | |
| >60 | 58(40.0) | 54(37.2) | |
| Gender | | | 0.524 |
| Male | 98(67.6) | 103(71.0) | |
| Female | 47(32.4) | 42(29.0) | |
| Comorbidity | | | 0.556 |
| No | 76(52.4) | 81(55.9) | |
| Yes | 69(47.6) | 64(44.1) | |
| Primary tumor side | | | 0.779 |
| Left-side colorectum | 111(76.6) | 113(77.9) | |
| Right-side colon | 34(23.4) | 32(22.1) | |
| TNM stage | | | |
| T stage | | | 0.158 |
| 1–3 | 73(50.3) | 61(42.1) | |
| 4 | 72(49.7) | 84(57.9) | |
| N stage | | | 0.583 |
| 0 | 59(40.7) | 62(42.8) | |
| 1 | 59(40.7) | 51(35.1) | |
| 2 | 27(18.6) | 32(22.1) | |
| IVTT[a] | | | 0.623 |
| Positive | 53(36.6) | 44(30.3) | |
| Negative | 73(50.3) | 69(47.6) | |
| Differentiation (primary) | | | 0.500 |
| Poor | 35(24.1) | 28(19.3) | |
| Moderate | 100(69.0) | 109(75.2) | |
| Well | 10(6.9) | 8(5.5) | |
| Metastases diameter (cm) | | | 0.857 |
| ≤3 | 128(88.3) | 127(87.6) | |
| >3, ≤5 | 17(11.7) | 18(12.4) | |
| Metastases number | | | 0.459 |
| 1 | 77(53.1) | 73(50.3) | |
| 2 | 30(20.7) | 40(27.6) | |
| 3 | 20(13.8) | 14(9.7) | |
| 4–5 | 18(12.4) | 18(12.4) | |
| Metastases distribution | | | 1.000 |
| Unilobar | 79(54.5) | 79(54.5) | |
| Multilobar (>1 lobe) | 66(45.5) | 66(45.5) | |
| Preoperative chemotherapy | | | 0.616 |
| No | 49(33.8) | 45(31.0) | |
| Yes | 96(66.2) | 100(69.0) | |

**Table 2** (*continued*)

| Characteristics | Resection ($n = 145$, %) | Ablation ($n = 145$, %) | *P* value |
|---|---|---|---|
| Postoperative chemotherapy | | | 0.860 |
| No | 18(12.4) | 19(13.1) | |
| Yes | 127(87.6) | 126(86.9) | |
| KRAS status[b] | | | 0.652 |
| Wild type | 41(28.3) | 44(30.3) | |
| Mutation type | 20(13.8) | 18(12.4) | |
| CEA level (ng/ml) | | | 0.724 |
| ≤5 | 65(44.8) | 68(46.9) | |
| >5 | 80(55.2) | 77(53.1) | |
| Timing of metastasis (months) | | | 1.000 |
| Synchronous (<12) | 18(12.4) | 18(12.4) | |
| Metachronous (≥12) | 127(87.6) | 127(87.6) | |

**Notes.**
[a] Data of 239 patients were available.
[b] Data of 123 patients were available.
PSM, Propensity score matching; IVTT, Intravascular tumor thrombus.

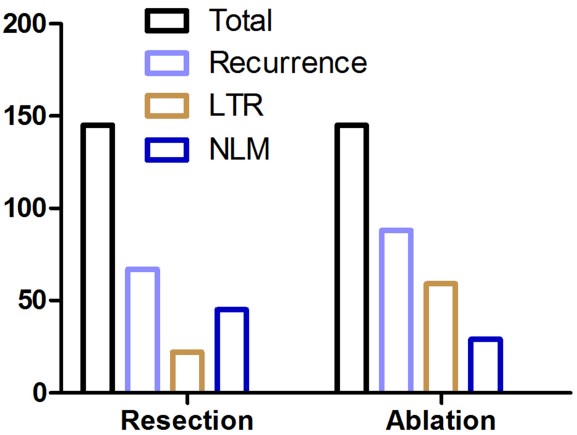

**Figure 3** **Recurrence parameters following resection and ablation.** *LTR* local tumor recurrence, *LNM* liver new metastasis.

## Complications

Death occurred in two patients of the resection group during hospitalization. Other complications, whose total incidence was 11.7%, included fistula ($n = 4$), peritonitis ($n = 3$), pleural effusion ($n = 2$), pneumothorax ($n = 2$), and subcapsular hematoma ($n = 2$) (Table 3). These patients received pertinent and appropriate treatment, and finally recovered.

## DISCUSSION

In this study, we investigated patients with colorectal metastases only limited in the liver, which were resectable-ablative. Our results showed that the presence or absence of liver

**Table 3  Recurrence and complication evaluation between resection and ablation.**

| | Resection (*n* = 145, %) | Ablation (*n* = 145, %) | *P* value |
|---|---|---|---|
| Recurrence Pattern | | | |
| Total (liver)[a] | 67(46.2) | 88(60.7) | 0.013[a] |
| LTR | 22(32.8)[b] | 59(67.0)[c] | <0.001 |
| LNM | 45(67.2) | 29(33.0) | |
| EHR only | 13(9.0) | 10(6.9) | |
| Early liver recurrence | 25(37.3)[d] | 24(27.3)[e] | 0.223 |
| Recurrence Number (liver) | | | |
| Mean | 2.06 ± 1.19 | 2.06 ± 1.26 | 0.637[f] |
| 1 | 31 | 43 | |
| 2 | 15 | 17 | |
| 3 | 7 | 8 | |
| >3 | 14 | 20 | |
| Complication | | | |
| Total | 19(13.1) | 15(10.3) | 0.465 |
| Peritonitis | 2 | 1 | |
| Pleural effusion | 1 | 1 | |
| Death | 2 | | |
| Chylous fistula | 3 | | |
| Biliary fistula | 1 | | |
| Intestinal obstruction | 2 | | |
| Liver function failure | 2 | | |
| Liver abscess | 1 | | |
| Cardiac insufficiency | 5 | | |
| Pneumothorax | | 2 | |
| Subcapsular hematoma | | 2 | |
| Infection | | 2 | |
| Hydropneumothorax | | 1 | |
| Pneumorrhagia | | 1 | |
| Thrombocytopenia | | 1 | |
| Empyrosis of the back skin | | 1 | |
| Right shoulder pain | | 1 | |
| Fever | | 1 | |
| Nausea | | 1 | |

**Notes.**
[a]Comparison between the two groups with and without liver recurrence was made by the $\chi^2$ test.
[b]Includes 8 patients with a combined LNM in the resection group.
[c]Includes 22 patients with a combined LNM in the ablation group.
[d]Data of 67 patients were available.
[e]Data of 88 patients were available.
[f]Comparison between the two groups was made by the Student *t* test.
LTR, local tumor recurrence; LNM, liver new metastasis; EHR, extrahepatic recurrence.

recurrence was one of the important factors affecting the survival duration, indicating the improvement in liver control was crucial to the prognosis and disease management.

The OS rates in the entire group and respective group at specific year in our study were comparable to the results of several other similar studies with a long-term cohort (*Abdalla*

**Table 4  Survival rates in the specific period.**

| Survival | OS | | LPFS | |
|---|---|---|---|---|
| | Resection (%) | Ablation (%) | Resection (%) | Ablation (%) |
| 1-year | 95.8 | 95.0 | 64.5 | 50.8 |
| 3-year | 69.8 | 60.1 | 49.4 | 29.5 |
| 5-year | 53.6 | 42.5 | 36.5 | 24.3 |
| 8-year | 45.1 | 32.9 | | |
| 10-year | 45.1 | – | | |

**Notes.**

OS, Overall survival; LPFS, Liver progression-free survival.

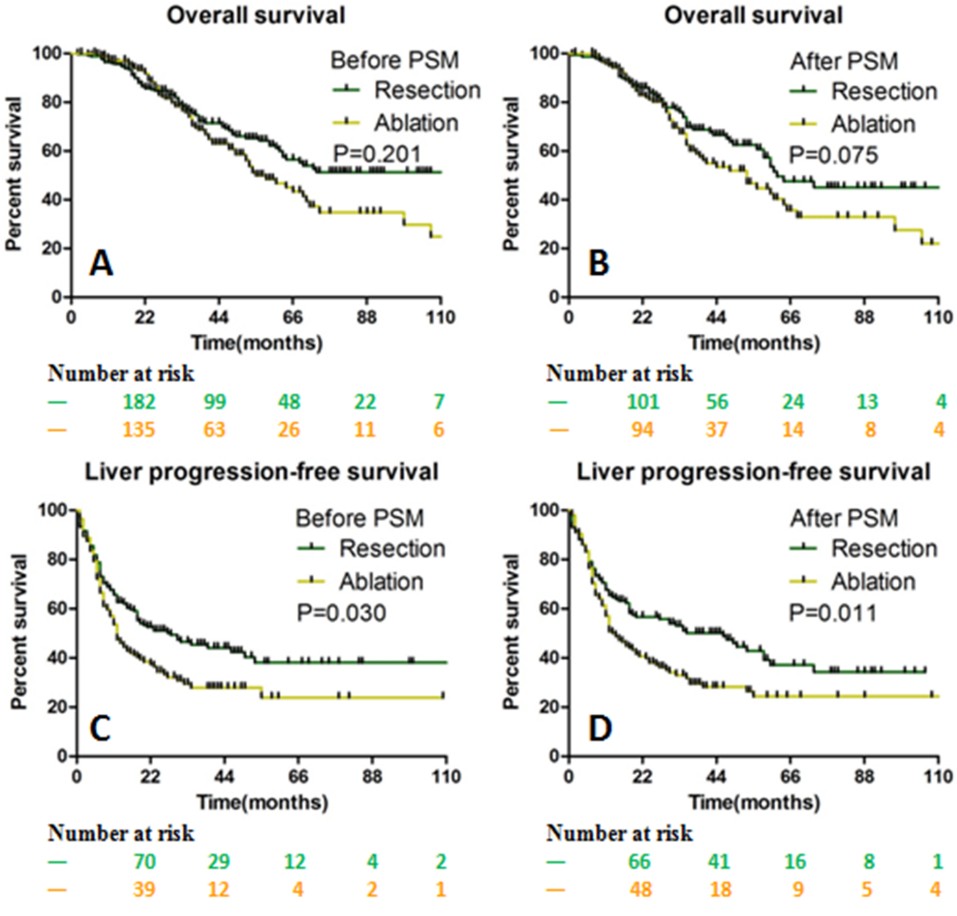

**Figure 4  The OS and LPFS curves for patients with CLOM.** OS between the resection and ablation groups were not significantly different before (A) and after (B) PSM. The LPFS between the resection and ablation groups was significantly different before (C) and after (D) PSM. *OS* Overall survival, *LPFS* Liver progression-free survival, *CLOM* Colorectal liver oligometastases, *PSM* Propensity score matching.

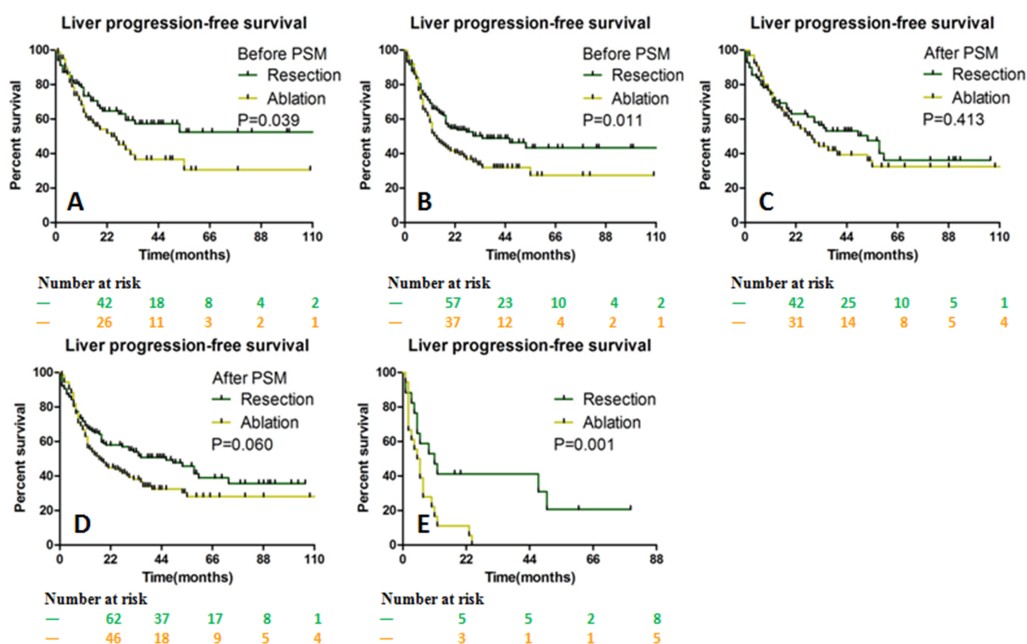

**Figure 5** **LPFS stratified by the size of liver metastases between the resection and ablation groups.** LPFS was significantly different before PSM for ≤ 3 cm solitary (A) and multiple lesions (B). After PSM, LPFS was not significantly different for ≤ 3 cm solitary (C) and multiple lesions (D), but was significantly longer in the resection group for > 3 cm lesion (E). *LPFS* Liver progression-free survival, *PSM* Propensity score matching.

*et al., 2004*; *Chua et al., 2011*; *Lee et al., 2015*; *Lin et al., 2018*; *Shady et al., 2016*). The LPFS rates at 1, 3 and 5 years in our ablation group were higher than those patients presented with recurrent (Van Tilborg et al. was about 42%–20%) (*Van Tilborg et al., 2011*), unresectable (*Dupre et al. (2017)* was about 20%-0) or extrahepatic disease at initial ablation (*Siperstein et al. (2007)* was about 35%–15%).

The LPFS before and after PSM in the resection group was longer than that in the ablation group, and the local tumor recurrence rate in the resection group was lower, indicating that resection had an advantage in lower recurrence. In the literature regarding ablation, although Solbiati et al. and Shady et al. reported that the 5- (31–47.8%) and 10-year (18%) survival rates were close to those after surgical resection, the local tumor recurrence rate was higher (40.4–54.0%). Also, the overall LPFS was shorter in patients treated with ablation compared to surgery, for both initial and recurrent lesions in many other studies (*Dupre et al., 2017*; *Sofocleous et al., 2011*; *Wang et al., 2018*). Recently, two non-inferiority prospective trials called LAVA (Liver resection surgery versus thermal ablation for colorectal Liver Metastases) and COLLISION (Colorectal liver metastases: surgery versus thermal ablation), respectively, are being carried out (*Gurusamy et al., 2018*; *Meijerink, Puijk & Van den Tol, 2019*). They are aimed on evaluating the treatment efficacy and benefit of thermal ablation compared with resection among patients with small lesions (0–3 cm) characterized by ablatable and resectable. The results are worthy of expectation and publication.

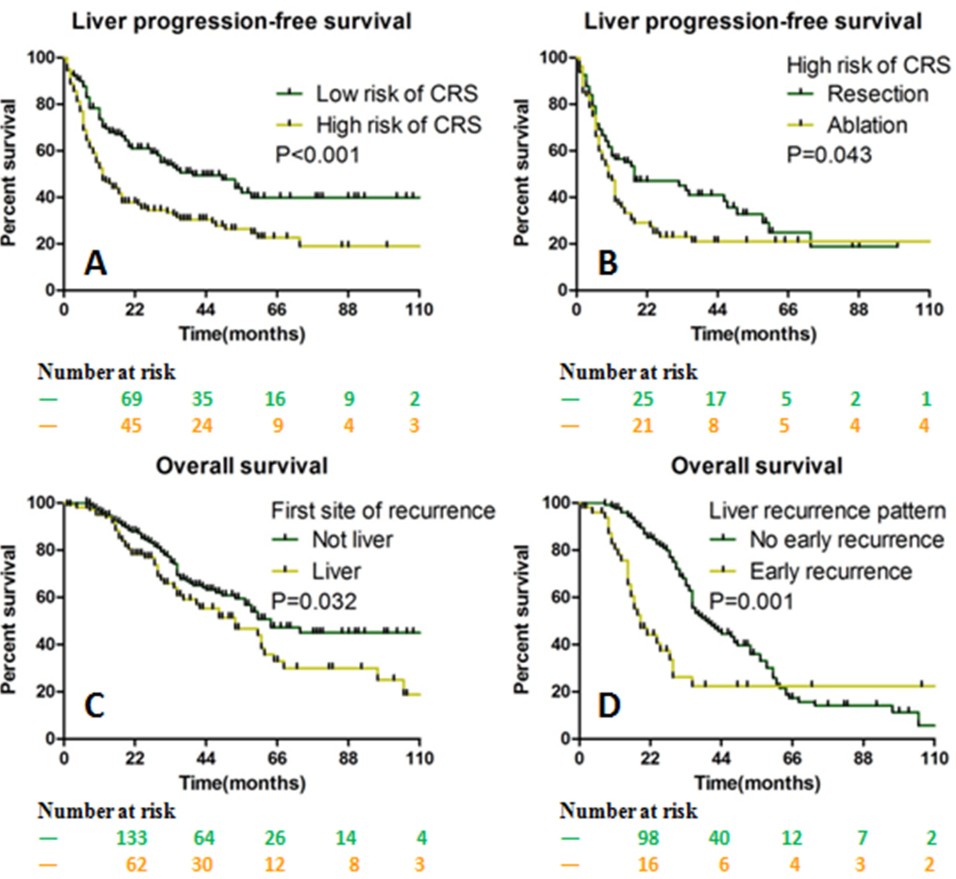

**Figure 6 The OS and LPFS stratified by CRS and liver recurrence parameters.** LPFS was longer in the low risk of CRS than in the high risk of CRS (A). For the subgroup of high risk of CRS, LPFS was longer in the resection group than in the ablation group (B). The OS was better in patients without liver recurrence than in those with liver recurrence (C). The OS was better in patients without early liver recurrence than in those with early liver recurrence (D). OS Overall survival, LPFS Liver progression-free survival, CRS Clinical risk score.

Recurrence is commonly seen in CLOM after curative treatment (*Lin et al., 2018*). The local tumor recurrence rate in our cohort was within the range of 43.6–57.3% from other long-time follow-up observations (*Otto et al., 2010*; *Park et al., 2008*), but higher than the results presented by *Lee et al. (2016)* and *Wang et al. (2018)* who showed rates of 29.2% and 40.4%, respectively. Additionally, the local tumor recurrence rate between the MWA and RFA was higher than the result of *Takahashi et al. (2018)*. The total case number and follow-up period in their studies were less and shorter than they were in our study, and the report from Lee et al. was focused on isolated hepatic metastasis. The resection group showed less liver recurrence at 1-, 3- and 5-year follow-up than the ablation group, which was in accordance with the results presented by *Abdalla et al. (2004)* and *Lee et al. (2015)*. We also noted that the liver recurrence pattern was different in the two groups. More local tumor recurrences were occurred in the ablation group, while liver new metastases were higher in the resection group, demonstrating that resection was superior in local disease control

**Table 5  Results of univariate and multivariate analyses of LPFS.**

| Variable | Univariate P | Multivariate HR (95% CI) | P |
|---|---|---|---|
| Age (≤60 years vs. >60 years) | 0.876 | | |
| Gender (male vs. female) | 0.064 | | |
| Primary tumor side (left vs. right) | 0.874 | | |
| Primary tumor location (colon vs. rectum) | 0.014 | | |
| T (1-3 vs. 4) | 0.005 | 1.511 (1.033–2.209) | 0.033 |
| N stage (negative vs. positive) | 0.003 | | |
| IVTT (negative vs. positive)[a] | 0.015 | | |
| Pathology | | | |
|    Poor | Reference | | |
|    Moderate | 0.072 | | |
|    Well | 0.463 | | |
| Metastases diameter (≤3 cm vs. >3 cm) | <0.001 | 2.174 (1.233–3.834) | 0.007 |
| Metastases number (1 vs. >1) | <0.001 | | |
| CEA before ablation (≤5 ng/ml vs. >5 ng/ml) | 0.003 | | |
| CEA after ablation (≤5 ng/ml vs. >5 ng/ml) | <0.001 | 1.950 (1.281–2.969) | 0.002 |
| Time of metastasis (≤12 months vs. >12 months) | 0.852 | | |
| Clinical risk score (low risk vs. high risk) | <0.001 | 1.706 (1.159–2.510) | 0.007 |
| KRAS status (wild vs. mutation)[b] | 0.495 | | |
| Treatment (resection vs. ablation) | 0.034 | | |
| Pre-procedure Chemotherapy (no vs. yes) | 0.463 | | |
| Post-procedure chemotherapy (no vs. yes) | <0.001 | | |

**Notes.**
[a] Data of 239 patients were available.
[b] Data of 123 patients were available LPFS Liver progression-free survival.
HR, Hazard ratio; IVTT, Intravascular tumor thrombus; CEA, Carcinoembryonic antigen.

(treated area) even when R0 ablation and resection were completely implemented, but the first site of recurrence in the liver was not associated with survival (*Siperstein et al., 2007*). We speculated there were several possible explanations for this result. First, heat injury might induce and stimulate inflammation, and the expression of angiogenesis molecular was increased, promoting tumor progression or metastasis after ablation (*Kumar et al., 2018*; *Ni et al., 2019*; *Zhang et al., 2017*). Second, R0 resection was confirmed by pathology, while complete ablation without microscopic validation. Although a multiposition biopsy was recommended during the ablation procedure, the tissue extracted from a local part in the ablation area might be partial and increase the risk of complications (*Sotirchos et al., 2016*). Third, the ablative tissue reacting to heat stimulation might present abnormal blood perfusion on contrast-enhanced images, occasionally influenced the determination of reactive hyperemia, and residual or viable tumor (*Chan et al., 2015*).

IVTT positive in the primary tumor at microscopy represented more obvious aggressive behavior with tumor growth, invasion, and metastasis than those with IVTT negative lesion (*Peng et al., 2019*). Similarly, when R0 resection or complete ablation was achieved, the LPFS in the ablation group was shorter than that in the resection group when IVTT was
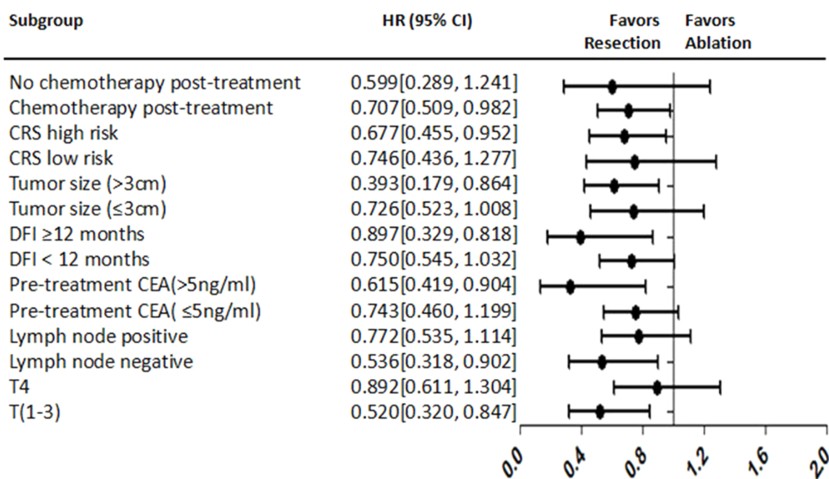

**Figure 7 Forest plot for LPFS.** *LPFS* Liver progression-free survival, *HR* Hazard ratio, *CRS* Clinical risk score, *DFI* Disease-free interval, *CEA* Carcinoembryonic antigen.

positive, although no statistical difference was shown. This suggested that in the high risk of recurrence condition, resection might still be recommended to IVTT positive patients due to its relatively lower recurrence risk compared with that of ablation.

CRS is a comprehensive index associated with prognosis (*Fong et al., 1999*). Tumor size is an important factor for the ablation effect (*Gillams & Lees, 2009*). As the size increases (>3 cm), the possibility of achieving complete ablation decreases and the risk of residual tumor increases. The modified CRS using a cutoff point of three cm instead of five cm for tumor size is optimal for evaluating the outcome between R0 resection and complete ablation (*Shady et al., 2016*). Our results showed that resection was beneficial for achieving longer LPFS in patients with high risk of CRS, although it was a poorer overall prognosis compared to patients with low risk of CRS.

With respect to ≤3 cm liver metastasis, liver disease control between the two groups was similar, but it was worse in the ablation group than that in the resection group when the liver lesion size was >3 cm, supporting the results described by Tilborg et al. and Lee et al. that resection and ablation provided similar outcome in the treatment of CLOM cases with hepatic tumor size ≤3 cm (*Lee et al., 2016*; *Van Tilborg et al., 2011*). This highlighted the finding that ablation provided equal benefit to surgery for small liver lesion as was presented in other researches, which showed that ablation was as effective as resection in the treatment of other small with/non solitary malignant tumors (*Chen et al., 2006*; *Uhlig et al., 2018*).

There are some limitations in our study. First, the retrospective nature and a limited number of patients from single center made our finding need to be validated in other prospective trials. Second, 131 patients were excluded from the final cohort after PSM, leading to raw data distortion. However, the baseline data before further analysis were more balanced; thus, avoiding amplification of a biased conclusion. Third, liver metastases in the ablation group were diagnosed on the basis of CEA and contrast enhancement of CT

or MRI, lacking of histopathologic evaluation of the target area, as some published reports had described that such an examination could identify important information about tumor biology and the potential risk for progression before or after treatment (*Motoyoshi et al., 2010*; *Sofocleous et al., 2013*). Moreover, cases with molecular characteristics such as KRAS and microsatellite test were limited in our cohort, and they might be helpful to predict and evaluate the prognostic outcome in these patients. Finally, we did not compare the difference in lesion location and depth in the liver as they were not included in the CRS analysis. Although some authors considered that location might not influence the oncologic outcome (*Chen et al., 2018*), the depth might affect the decision on resection or ablation, and treatment efficacy, as the deeper the lesion, the higher the risk and the more the difficulty for complete eradication of the lesion.

## CONCLUSION

This retrospective study investigating the outcome in CLOM patients with liver lesion number ≤5 and size ≤5 cm, which was resectable-ablative, indicated that resection could result in longer liver recurrence-free survival than ablation when the size was >3 cm or there was a high risk of CRS. But for ≤3 cm liver metastases, their treatment efficacies were comparable.

### Funding

This work was supported by grants from the National Natural Science Foundation of China (No. 81871467, 81571785, 81771957) and the National Key Research and Development Program of China (No. 2017YFA0205200). The funders had no role in study design, data collection and analysis, decision to publish, or preparation of the manuscript.

### Grant Disclosures

The following grant information was disclosed by the authors:
National Natural Science Foundation of China: 81871467, 81571785, 81771957.
National Key Research and Development Program of China: No. 2017YFA0205200.

### Competing Interests

The authors declare there are no competing interests.

### Author Contributions

- Ma Luo performed the experiments, prepared figures and/or tables, authored or reviewed drafts of the paper, and approved the final draft.
- Si-Liang Chen performed the experiments, prepared figures and/or tables, software, and approved the final draft.
- Jiawen Chen analyzed the data, prepared figures and/or tables, data collection, and approved the final draft.
- Huzheng Yan analyzed the data, authored or reviewed drafts of the paper, and approved the final draft.

- Zhenkang Qiu and Guanyu Chen analyzed the data, prepared figures and/or tables, and approved the final draft.
- Ligong Lu and Fujun Zhang conceived and designed the experiments, authored or reviewed drafts of the paper, and approved the final draft.

## Human Ethics

The following information was supplied relating to ethical approvals (i.e., approving body and any reference numbers):

The Sun Yat-sen University Cancer Center granted Ethical approval to carry out the study within its facilities (approval number: 20170302015).

## Data Availability

Raw data is available as a Supplemental File.

## Supplemental Information

Supplemental information for this article can be found online at http://dx.doi.org/10.7717/peerj.8398#supplemental-information.

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
