# Peer review of "Resection vs. ablation for lesions characterized as resectable-ablative within the colorectal liver oligometastases criteria: a propensity score matching from retrospective study"

_PeerJ, doi:10.7717/peerj.8398_

## Round 0.1 · original submission · Minor Revisions

Your manuscript has been reviewed and requires modifications prior to making a decision. The comments of the reviewers are included at the bottom of this letter. Reviewer 1 also indicated that the manuscript needs extensive English editing. I agree with this evaluation and I would, therefore, request for the manuscript to be revised accordingly. I would also like to suggest the following changes:

• Correct “SPSS” to “IBM SPSS”.
• Provide percentages in Tables 1-2-3.
• Merge survival figures as one figure, because the total number of figures (22 figures) are really high.

·

Basic reporting

English may need some small correction. Background, article structure and reported data are well written.
Figures are very explicative

Experimental design

Authors reported and interesting propensity score matching analysis on patients affected by liver oligometastases from colorectal cancer treated with surgery, or MWA and RFA.
The subject of the article is of primary importance since the oligometastatic disease is showing increasing interest in the scientific community and exploring the role of more conservative treatment is of primary importance.
The aim of the study is well presented and the statistical analysis was correctly performed
Patients and treatment characteristics are well presented.

Validity of the findings

Results of the study are very interesting and confirms what was derived from retrospective series. The most interesting analysis regards smaller lesions, for which local and more tolerable treatments provided the same local control. )To be noted 2 deaths occurred in the surgery group)
Conclusions are well stated

Additional comments

No specific comments

Reviewer 2 ·

Basic reporting

The authors compared a retrospective serie of patients treated either by surgical resection or focal destruction using radiofrequency for colorectal liver metastasis. To limit the inconvenience of retrospective analysis and to avoid bias, they used a propensity score matching. So that, 145 patients were included in each group.

Experimental design

The main result was that when liver metastases were up to 3 cm or in case of high CRS, surgical resection obtained better survival than ablative treatment. Otherwise, when metastases were < 3cm, treatments efficacies were similar.

Validity of the findings

This is an interesting article that try answering an unsolve problem. Authors used a propensity score to increased comparability of the two groups.
I have few minor comments to do :
- authors did not differenciate radiofrequency ablation (RFA) for microwave ablation (MWA) while a study from Takahashi et al. published in HPB in 2018 showed that there is 10% of local recurrence after MWA and 20% after RF (p=0.02).
- this study has been performed because "there has been no prospective or retrospective study reporting the comparison between surgery and ablation". That is right but... the LAVA study is currently being included and it could be interesting to add a reference about this trial : Liver resection surgery versus thermal ablation for colorectal LiVer MetAstases (LAVA): study protocol for a randomised controlled trial. Gurusamy and al. Trials 2018.

Additional comments

No comment

Reviewer 3 ·

Basic reporting

The article is genrally well written, and sufficient data, tables and figures were included.

Experimental design

No comments

Validity of the findings

Novelty is limited, although conclusions are valid.

Additional comments

- I recomment to minimalize the use of abbreviations, which impair the readability of the paper.
- Provide the numbers at riks in the Kaplan-Meier figures.
- Clearly sate in the methods section how an event van defined (recurrence and death?)
- Provide percentages in the tables and not only numbers of patients (e.g. table 1)

---

## Round 0.2 · accepted · Accept

The authors addressed the reviewers' concerns and substantially improved the content of MS. So, based on my own assessment as an editor, no further revisions are required and the MS can be accepted in its current form.

·

Basic reporting

Authors reviewed paper properly

Experimental design

No comments

Validity of the findings

Article report an important analysis of alternative treatment option for liver metastases and useful predictive factors for treatment personalization

Additional comments

Author answer properly to reviewer comments

Reviewer 3 ·

Basic reporting

The authors have correctly addressed my comments.

Experimental design

No new comments on this topic.

Validity of the findings

Analysis are appropriate, and comclusions adhere to the results.

Additional comments

Authors have adressed the comments. No further comments.